# The Relationship between Effort-Reward Imbalance for Learning and Academic Burnout in Junior High School: A Moderated Mediation Model

**DOI:** 10.3390/bs13010028

**Published:** 2022-12-28

**Authors:** Yuanru Wang, Yidan Gao, Xiaoyin Zhang, Jingyi Shen, Qiangqiang Wang, Yingjie Wang

**Affiliations:** 1School of Teacher Education, Huzhou University, Huzhou 313000, China; 2The Key Laboratory of Brain Science and Children’s Learning of Huzhou, Huzhou University, Huzhou 313000, China

**Keywords:** Effort-Reward Imbalance for Learning, learning satisfaction, academic burnout, resilience, junior high school students

## Abstract

Although effort-reward imbalance has been proven to affect academic burnout, how effort-reward imbalance affects academic burnout remains unclear. This study, from the perspective of learning satisfaction and resilience, investigates how effort-reward imbalance affects academic burnout and reveals the influence of effort-reward imbalance on academic burnout. A sample of 755 junior high school students was assessed using the Revised Effort-Reward Imbalance for Learning Scale, Revised Learning Satisfaction Scale, Academic Burnout Scale, and Resilience Scale. Junior high school students’ effort-reward imbalance rates for learning, learning satisfaction, and academic burnout were all significantly correlated with each other; learning satisfaction mediated the relationship between them. Learning satisfaction mediated the relationship between junior high school students’ effort-reward imbalance rate for learning and academic burnout, and resilience negatively moderated the path from junior high school students’ effort-reward imbalance rate from learning to learning satisfaction. The results suggest that improving students’ resilience can effectively decrease the negative effects of effort-reward imbalance.

## 1. Introduction

Academic burnout results in negative psychology towards classmates, school, and studying, and a distant attitude can result from long-term academic burden and pressure, including two emotional exhaustion, low personal achievement, and cynicism [1]. Junior high school is a critical period for students’ growth and development, and their cognition and self-awareness of the world are still developing. Under long-term academic burnout, junior high school students will probably have psychological symptoms such as depression and anxiety [2], and a series of externalised problems such as truancy or dropping out [3,4]. During the junior high school years, if students’ academic burnout is not handled in a timely and effective manner, it will affect their studies and lives and may cause unhealthy emotions such as learning weariness and depression. These emotions can seriously affect their physical and mental health. Junior high school students are the hope of the country, and their physical and mental health exert a vital influence on individual development as well as the development of society [5]. Since academic burnout has a crucial impact on the growth of junior high school students, to prevent or reduce the level of academic burnout of junior high school students, scholars have conducted in-depth research on the factors and mechanisms of academic burnout from the aspects of academic pressure [6], perceived teachers’ emotional support [7], stressors [8], and effort-reward imbalance [9]. Research on the relationship between effort-reward imbalance and academic burnout found that effort-reward imbalance can significantly positively predict students’ academic burnout.

The effort-reward imbalance (ERI) model [10] is a classical work stress theory proposed by German physiologist Siegrist. The model argues that a lack of reciprocity between the amount of time and effort an individual puts into a job and the amount of money, respect, recognition, and career opportunities that an individual obtains from a job (such as high effort and low reward) is the effort-reward ratio, which leads to negative emotions and an autonomic nervous system affected by continuous tense reactions [11], at the same time increasing the risk and prevalence of insomnia [12,13], and mental health issues [14,15]. With the deepening and transfer of research, the theory has been applied to the field of education, including the two dimensions of effort and reward. Learning effort is the time and energy students pay for learning; learning reward is the rewards (grades and returns, etc.) students achieve through learning [9]. Although previous studies have confirmed that effort-reward imbalance can affect students’ academic burnout, how the effort-reward imbalance affects student academic burnout remains unclear. Burnout is caused by the exhaustion of internal resources (emotion) and the erosion of cognitive and physiological resources [16]. According to the job demand–job resource model theory [17], the imbalance between job demands and job resources leads to excessive work pressure and then to job burnout [18], indicating that job pressure is the direct factor causing job burnout (Brill, 1984). Based on the work stress model of effort-reward imbalance, individuals with effort-reward imbalance are prone to energy consumption and emotional exhaustion. That is, the effort-reward imbalance may lead to emotional exhaustion caused by too much stress at work, which contributes to burnout. Academic burnout, as one type of burnout, is the state of energy exhaustion caused by students coping with academic pressure, which arises from an excessive academic burden in the cognitive development stage [5]. Studies have found that individuals in a state of long-term effort-reward imbalance (high effort and low return) will increase the risk of negative emotions, and even lead to emotional exhaustion [19,20], resulting in burnout. Studies have shown that effort-reward imbalance easily produces academic pressure [21], and the generation of academic burnout is an important reflection of academic pressure. Therefore, effort-reward imbalance may affect academic burnout.

Learning satisfaction is an important research topic in positive psychology and an important index for measuring students’ mental health. Learning satisfaction is transformed from “customer satisfaction” [22], which refers to the psychological feeling of realizing the needs and wishes of individuals when they participate in learning activities, and the degree to which the needs and wishes are satisfied in the learning process [23]. If the need or desire is met, learning satisfaction is high; if the need or desire is not met, learning satisfaction is low [24]. The stress model of effort-reward imbalance posits that individuals need to make an effort to obtain returns. When effort and reward are out of balance, that is, the individual’s effort does not receive a corresponding return, a stress response will occur; that is, the effort-reward imbalance is a stressor [10]. Stress is a physical or psychological experience that mainly manifests as a kind of maladaptation or gap, that is, the gap between psychological expectations and reality. Moreover, the academic pressure of middle school students directly affects their learning satisfaction [25]. Therefore, effort-reward imbalance may affect learning satisfaction. According to the study, high learning satisfaction can help students invest more energy in learning and buffer or even offset the negative effect of academic burnout. In contrast, it is easy to fall into the development dilemma of emotional exhaustion and a low sense of achievement [26]. According to Spence and Evans, when an individual is dissatisfied, his or her learning motivation, learning interest, and learning behaviour may be reduced [27], which easily leads to burnout. Therefore, learning satisfaction may affect academic burnout.

When the phenomenon of effort-reward imbalance occurs in students’ learning processes, the generation of an effort-reward ratio not only directly reduces students’ learning satisfaction but also causes students’ academic burnout through the decline of learning satisfaction. Therefore, hypothesis 1 can be deduced.

**Hypothesis** **1.**
*Learning satisfaction mediates the relationship between effort-reward imbalance and academic burnout.*


Resilience is the ability to recover from negative experiences and flexibly adapt to the changing environment without being affected by severe stress and adversity in terms of psychological function and development (Lazarus 1993; Werner 1993) [28]. Research has found that individuals with high resilience have positive self-concept and cognition [29], and high flexibility in the selection of response in uncertain events [30], which helps them maintain healthy emotions, enhance their endurance and self-efficacy to cope with setbacks [31], and adopt positive strategies to cope with trauma and corresponding emotional experiences [32]. Teenagers who chose proactive coping method showed less emotional and behavior problems than those who used avoidant coping method [33] These skills can effectively reduce psychological behaviour problems in adolescence [34] to guarantee the normal functioning of individuals after trauma [35]. Although the effort-reward ratio reduces the learning satisfaction of individuals and increases students’ academic burnout, according to the cognitive basis model of resilience, it can be speculated that a high level of resilience can effectively ease the impact of the effort-reward ratio on learning satisfaction and then reduce students’ academic burnout, even for students with an effort-reward ratio in learning. In contrast, when an individual’s resilience is low, low resilience is not enough to buffer the impact of the effort-reward ratio on learning satisfaction and thus increases academic burnout. Therefore, hypothesis 2 can be deduced.

**Hypothesis** **2.***Resilience is likely to moderate the relationship between effort-reward imbalance and learning satisfaction*.

Through the above analysis, this study proposes the following model to describe how effort-reward imbalance affects academic burnout based on the effort-reward imbalance model, job demand-job resource model, and pay-to-reward imbalance stress model. Figure 1 shows the proposed theoretical model.

## 2. Methods

### 2.1. Participants and Procedures

The present study recruited 972 students from Grades 1 to 3 in several middle schools in Gansu Province, China, by cluster sampling. After excluding missing data, 755 completed questionnaires were obtained, and the acceptable rate was 77.67%. Boys and girls accounted for 280 and 475, respectively. Additionally, the participants were from three grades: 426 students in junior grade one (%), 321 in junior grade two (%), and 8 in junior grade three (%). We obtained the participants’ informed consent prior to administering the questionnaire, and the research protocol was approved by the medical ethics committee of Huzhou University.

### 2.2. Measures

Effort-Reward Imbalance for Learning Scale. We adopted two subscales, effort (3 items) and reward (4 items), from the Effort-Reward Imbalance for Learning Scale prepared by Fukuda and Ya Mano and revised by Chu K [36]. The questionnaire uses a two-point Likert scale, with 1 indicating “no” and 2 indicating “yes”. For example, students are asked to give “yes” or “no” answers to “When I am studying in school or in a class, I often have to stop studying because of other people’s interference” and “I try to do well in class”. LERI ratio = effort score/(reward scores × C), where C is the adjustment coefficient (the ratio of the number of items in the effort dimension to the number of items in the reward dimension). Generally, C = 0.75 and an effort-reward imbalance rate > 1 indicate effort-reward imbalance—in other words, high effort and low reward. An effort-reward imbalance ratio ≤ 1 indicates that there is no imbalance between effort and reward. This scale has been widely used in related studies [9] and demonstrates good reliability and validity.

Resilience Scale. We adopted the Resilience Scale for Adolescents prepared by Hu Y and Gan Y [37]. The scale includes five dimensions: goal focus (4 items), emotional control (6 items), positive cognition (4 items), family support (6 items), and interpersonal assistance (6 items). The subjects were asked to judge the extent to which the given question was in accordance with their actual situation. The questionnaire uses a five-point Likert scale, ranging from “strongly disagree” (1) to “strongly agree” (5). The higher the score is, the better the resilience of the individual. For example, students were asked to judge, on a scale of 1 to 5, how well “failure always discourages me” and “my parents respect my opinion” matched their actual situation. This scale has been widely used in related studies [38] and demonstrates good reliability and validity. In our present research sample, Cronbach’s alpha coefficient was 0.86.

Learning Satisfaction Scale. We adopted the Learning Satisfaction Scale for middle school students created by Zhang Qing. The scale includes five dimensions: teacher teaching, the results of experience, the school environment, parent respect, and peer assistance. It has a total of 26 questions, such as “The teacher’s teaching method is very efficient” and “I like our headteacher very much”. The questionnaire uses a four-point Likert scale, ranging from “strongly disagree” (1) to “strongly agree” (4). The higher the score is, the higher the individual’s learning satisfaction. This scale has been widely used in related studies [11] and demonstrates good reliability and validity. In our present research sample, Cronbach’s alpha coefficient was 0.95.

Academic Burnout Scale. We adopted the Academic Burnout Scale for middle school students prepared by Wu Y, Dai X, Wen Z, and Cui H [39]. The questionnaire includes three dimensions: physical and mental exhaustion (4 items), academic alienation (5 items), and efficacy (7 items). Examples include “I study energetically” and “I study so badly that I want to give up”. The questionnaire uses a five-point Likert scale, ranging from “strongly disagree” (1) to “strongly agree” (5). The higher the score is, the higher the degree of an individual’s academic burnout. This scale has been widely used in related studies [40] and demonstrates good reliability and validity. In our present research sample, Cronbach’s alpha coefficient was 0.87.

### 2.3. Statistical Analysis Tool

SPSS 26.0 and the process plug-in were used to analyse the collected data, and the Pearson correlation method was used to analyse the correlation between the four core variables. The bootstrap method was used to test the mediating effect. Samples were set 5000 times with a 95% confidence interval.

## 3. Results

To exclude the common method bias caused by the questionnaire method, the Harman single-factor test was used. Exploratory factor analysis was conducted on all items of the LERIS, Resilience Scale, Learning Satisfaction Scale for Middle School Students, and Academic Burnout Scale for Middle School Students. The results of unrotated factor analysis showed that the variation explained by the first factor is 23.80%, which is far less than the critical value of 40%. Therefore, it can be determined that there is no total variance explained.

### 3.1. Correlation Analysis of Variables and Demographic Differences

The correlation analysis of each variable showed that the effort-reward imbalance rate was negatively correlated with resilience and learning satisfaction and positively correlated with academic burnout. Resilience is positively correlated with learning satisfaction and negatively correlated with academic burnout. Learning satisfaction is negatively correlated with academic burnout. See Table 1 for details.

### 3.2. Mediating Effect of Learning Satisfaction in the Relationship between Effort-Reward Imbalance and Academic Burnout

Model 4 was selected from the PROCESS plug-in of SPSS 26.0, and the bootstrap method proposed by Hayes was used to verify the mediating effect of learning satisfaction on effort-reward imbalance and academic burnout (Table 2). After controlling for gender and grade, the results showed that effort-reward imbalance significantly negatively predicted learning satisfaction (β = −0.743, *p* < 0.01) and positively predicted academic burnout (β = 0.927, *p* < 0.01). When effort-reward imbalance and learning satisfaction together predicted academic burnout, learning satisfaction significantly negatively predicted academic burnout (β = −0.657, *p* < 0.01), and effort-reward imbalance still played a significant positive predictive role in academic burnout (β = 0.438, *p* < 0.01). The preliminary results indicate that learning satisfaction mediates the relationship between the disequilibrium of effort and reward and academic burnout.

Table 3 and Figure 2, with effort-reward rate as the independent variable, academic burnout as the dependent variable, and learning satisfaction as the mediating variable, was used to investigate the relationship between effort-reward imbalance and academic burnout, including the direct effect and the indirect effect with learning satisfaction as the mediating variable.

The results showed that the direct effect of the effort-reward rate on academic burnout was 0.4381, 95% CI (0.2972, 0.5791), and the indirect effect mediated by learning satisfaction was 0.4887, 95% CI (0.3869, 0.6019). Based on the above data, the confidence intervals of the direct and indirect effects do not contain zero. This indicates that the indirect effect of effort-reward imbalance on academic burnout is statistically significant. After controlling for learning satisfaction, the direct effect of effort-reward imbalance on academic burnout is still significant. Therefore, learning satisfaction is a partial intermediary between effort-reward imbalance and academic burnout, and the intermediary accounts for 52.72%.

### 3.3. The Moderating Effect of Resilience in the Relationship between Effort-Reward Imbalance and Learning Satisfaction

Model 7 in the PROCESS plug-in of SPSS 26.0 and the bootstrap method proposed by Hayes were used to test the moderating effect of resilience. The effort-reward rate was taken as the independent variable, academic burnout as the dependent variable, learning satisfaction as the mediating variable, resilience as the moderating variable, and grade and gender as the control variables.

The results showed (Table 4 and Table 5) that the moderated mediation model was significant (R^2^ = 0.3780, F = 91.0244, *p* < 0.01). Specifically, the effort-reward rate played a significant positive predictive role in academic burnout, and the interaction between effort-reward imbalance and resilience played a negative predictive role in learning satisfaction. This indicates that the path of effort-reward imbalance and learning satisfaction is moderated by resilience.

In addition, in order to explain the moderating effect of resilience more specifically, a simple slope test was used to analyse the moderating effect between the need to satisfy resilience and the effort-reward imbalance and academic burnout. After all variables were standardized, the sample size of 5000 was selected by Bootstrap method, and the confidence interval excluding 0 meant that the parameters were significant. The moderated mediation effect analysis was carried out with model 7. A simple slope plot is drawn based on the values of learning satisfaction corresponding to high or low resilience and payback imbalance. 

Effort-reward imbalance plays a negative predictive role in learning satisfaction, indicating that with a reduction in resilience level, the predictive effect of the effort-reward rate on learning satisfaction gradually decreases (Figure 3).

## 4. Discussion

Considering that it is not clear how effort-reward imbalance affects students’ academic burnout, this study investigates this question from the perspective of learning satisfaction and resilience based on the effort-reward imbalance model, job demand–job resource model, and effort-reward imbalance stress model. The results revealed that effort-reward imbalance significantly and positively predicts academic burnout, which is consistent with previous research results [9,40]. The reason for this relationship is that when the effort-reward imbalance circumstance appears in the learning process of students—in other words, high effort and low rewards—for individuals who are prone to negative emotional reactions [41], alienating attitudes, and many negative psychological symptoms [1], such as anxiety [42] and depression [43], this ultimately leads to increased academic burnout [40]. Therefore, the effort-reward rate can positively predict the state of academic burnout. When individuals pay-reward imbalance , it is easy for their emotional stability to be reduced, resulting in academic burnout.

### 4.1. The Mediating Role of Learning Satisfaction

The results revealed that learning satisfaction partly mediates the relationship between effort-reward imbalance and academic burnout. That is, effort-reward imbalance can both directly affect academic burnout and indirectly affect it through learning satisfaction. On the one hand, effort-reward imbalance significantly negatively predicts learning satisfaction. It is an important task for middle school students to acquire relevant knowledge through learning and promote their cognitive, emotional, and social abilities and qualities. The occurrence of effort-reward imbalance means that students’ efforts in learning activities do not yield a corresponding return, which is a significant negative life event. Learning satisfaction is the degree of academic satisfaction, and negative life events have a direct impact on life satisfaction [44]. Therefore, this study finds that effort-reward imbalance negatively affects learning satisfaction. On the other hand, learning satisfaction also significantly and negatively predicts academic burnout. As an indicator of learning stress, learning satisfaction is an important factor leading to students’ academic burnout. When students are not satisfied with learning activities for various reasons, they experience negative weariness, loss of confidence in learning, and so on, which leads to academic burnout.

Learning satisfaction is a positive psychological state that has a buffer effect on the relationship between effort-reward imbalance and academic burnout. While directly leading to academic burnout [9,40], effort-reward imbalance also affects students’ drive, learning behaviours, and learning satisfaction [25] by influencing their sense of feedback in the learning process and then indirectly affects their sense of academic burnout [45]. The result verifies the hypothesis 1 reveals the mediating role of learning satisfaction in the relationship between effort-reward imbalance and academic burnout and clarifies an important path for the influence of pay-off imbalance on middle school students’ academic burnout, which helps deepen the understanding of the mechanism between effort-reward imbalance and academic burnout.

### 4.2. The Moderating Effect of Resilience

The results revealed that resilience moderates the effect of effort-reward imbalance on learning satisfaction. According to the stress buffer model, when individuals face pressure, social support decreases the pressure and then has a positive impact on individuals to protect their physical and mental health [46]. This study found that resilience can decrease the negative impact of effort-reward imbalance on learning satisfaction, and the research results further support the stress buffer model. As a positive psychological trait, resilience can moderate individual protective resources and help individuals cope well in adverse situations [47,48]. Kumpfer [49] thinks that the development of psychological resilience is realized in the dynamic process of the simultaneous action of negative life events such as pressure and adversity and protective factors. This means that when experiencing a negative life event, such as effort-reward imbalance, individuals moderate their psychological resilience to resist, weaken the impact, and protect their positive psychological traits. In addition, the effort-reward rate may mitigate the positive psychological trait of individual resilience, reduce the protection capabilities of resilience, and lead to an increase in the negative impact of the effort-reward rate on individual learning satisfaction. Therefore, the effort-reward imbalance of students’ studies will have a negative effect on their learning satisfaction. In addition, individuals with a high level of resilience have higher flexibility [33] and are better at coping with adverse life events and maintaining a positive emotional state, which can buffer the negative effect of effort-reward imbalance on learning satisfaction. This study suggests that improving students’ resilience levels can decrease the negative impact of effort-reward imbalance on learning satisfaction [33]. Obviously, the results verify that hypothesis 2 reveal that resilience influences academic burnout by regulating the pathway of effort-reward imbalance and learning satisfaction, which helps deepen the understanding of the mechanism between effort-reward imbalance and academic burnout.

The findings of our study have several implications for interventions for academic burnout. First, effort-reward imbalance occurs when the time and energy invested do not yield a corresponding return [50]. Based on the perspective of students’ studies, it is believed that the effort-reward imbalance can be reduced from external evaluation standard adjustment and reasonable internal attribution. Second, the mediating effect of learning satisfaction suggests that teachers’ teaching, school learning systems and facilities, and family and social support can be used to intervene in academic burnout. Third, the moderating effect of psychological resilience suggests that individual resilience should be enhanced.

How the effort-reward imbalance affects students’ academic burnout remains unclear. Based on the effort-reward imbalance model, job demand–job resource model and effort-reward imbalance stress model, we put forward two hypotheses to answer how the effort-reward imbalance affects students’ academic burnout. Through the investigation of 755 junior high school students, it is found that satisfaction plays a mediating role between effort-reward imbalance and academic burnout. Moreover, the influence of effort-reward imbalance on academic burnout is moderated by resilience through the mediating effect of satisfaction. This study verifies our hypothesis and reveals the mechanism of the influence of effort-reward imbalance on academic burnout from the perspective of resilience and satisfaction and has enriched the understanding of middle school students’ effort-reward imbalance and academic burnout and provide a certain extent of enlightenment and reference value for enhancing students’ learning satisfaction and reducing students’ academic burnout.

### 4.3. Limitation

Although this study is based on the effort-reward imbalance model, job demand–job resource model, and effort-reward imbalance stress model, it examines the mechanism of effort-reward imbalance affecting middle school students’ academic burnout from the perspective of learning satisfaction and resilience and enriches and improves the research on the relationship between effort-reward imbalance and academic burnout. However, this study still has several limitations to be further addressed. First, in addition to affecting students’ learning satisfaction, does effort-reward imbalance affect students’ academic burnout in other ways? If so, what is the mechanism of its influence? Second, this study found that the resilience of students can decrease the negative impact of return on effort-reward imbalance on learning satisfaction and then decrease the impact of effort-reward imbalance on middle school students’ academic burnout behaviours. In fact, in addition to students’ resilience, factors such as students’ personality characteristics, education method, and support of parents and teachers could theoretically decrease the impact of the effort-reward imbalance on learning satisfaction. When many factors, such as students’ resilience, personality characteristics, education method, and support of parents and teachers, are interwoven, how does resilience affect the influence of the effort-reward imbalance on learning satisfaction and then decrease the influence of students’ academic burnout behaviours? Among actual students, the effort-reward imbalance will often interweave with many factors to affect students’ academic burnout. An in-depth study of the above questions can further increase our understanding of how effort-reward imbalance affects students’ academic burnout and guide students’ learning activities more effectively. Due to limited time and energy, this study failed to collect more variable information to investigate the deep relationship between effort-reward imbalance and academic burnout in a more comprehensive way. It is hoped that a follow-up series of studies can further investigate the above problems and reveal the internal mechanism by which effort-reward imbalance affects academic burnout.

## 5. Conclusions

The following conclusions can be drawn from this study: (1) Learning satisfaction plays a partial mediating role in the relationship between effort-reward imbalance and academic burnout; and (2) Resilience has a significant moderating effect on the relationship between effort-reward imbalance and learning satisfaction. The conclusions implying that improving students’ resilience can decrease the direct impact of effort-reward imbalance on students’ academic satisfaction and then decrease the impact of the decline in learning satisfaction caused by effort-reward imbalance on middle school students’ academic burnout.

## Figures and Tables

**Figure 1 behavsci-13-00028-f001:**
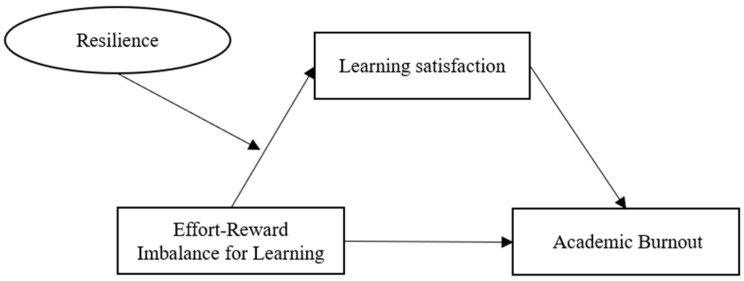
Proposed model.

**Figure 2 behavsci-13-00028-f002:**
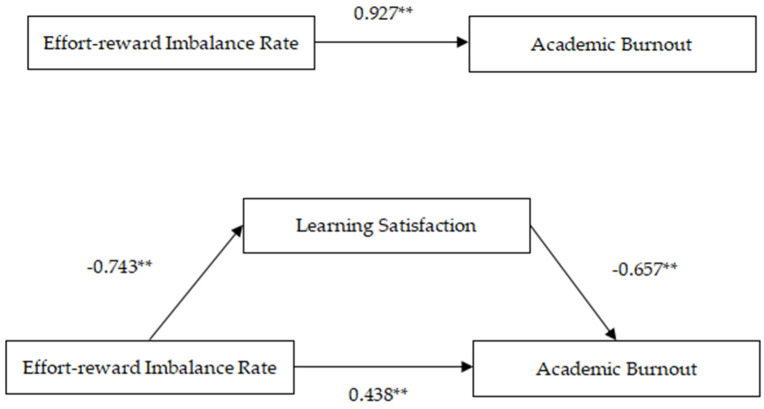
Mediating model of learning satisfaction, ** indicates *p* < 0.01.

**Figure 3 behavsci-13-00028-f003:**
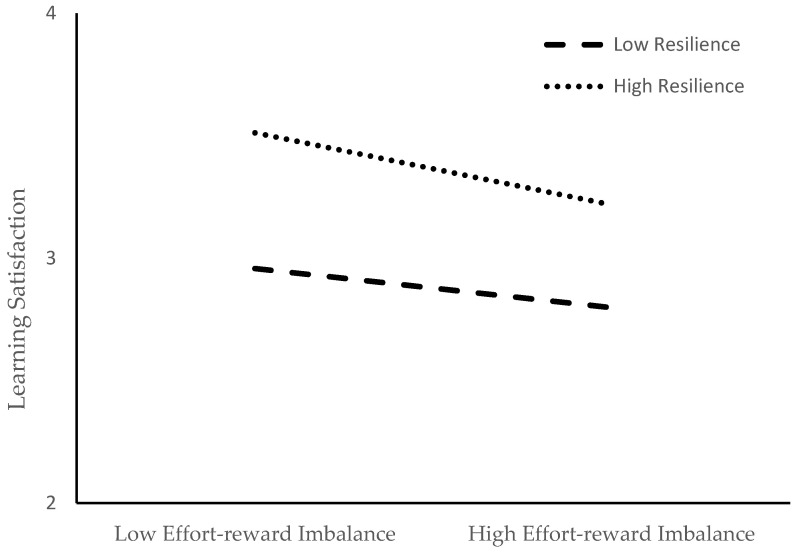
Moderating effect of resilience between effort-reward imbalance and learning satisfaction.

**Table 1 behavsci-13-00028-t001:** Average, standard deviation, and correlation coefficient of each variable.

	1	2	3	4	5
Effort-reward Imbalance Rate	1.00				
Resilience	−0.43 **	1.00			
Learning Satisfaction	−0.40 **	0.54 **	1.00		
Academic Burnout	0.40 **	−0.66 **	−0.61 **	1.00	
Grade	0.12 **	−0.07	−0.26 **	0.06	1.00
Mean	1.18	3.35	3.14	2.68	1.45
S. D.	0.27	0.61	0.54	0.64	0.52

Note: ** *p* < 0.001; all values are rounded to two decimal places.

**Table 2 behavsci-13-00028-t002:** Mediating model test of learning satisfaction.

Result Variables	Predictors	R	R^2^	F	β	LLCI	ULCI	t
Academic Burnout	Effort-Reward Imbalance Rate	0.406	0.165	49.518	0.927	0.773	1.081	11.796 **
Learning Satisfaction	Effort-Reward Imbalance Rate	0.457	0.209	66.010	−0.743	−0.870	−0.617	−11.524 **
Academic Burnout	Effort-Reward Imbalance Rate			129.425	0.438	0.297	0.579	6.102 **
	Learning Satisfaction	0.639	0.408		−0.657	−0.731	−0.584	−17.560 **

Note: ** indicates *p* < 0.01.

**Table 3 behavsci-13-00028-t003:** Mediating effect of learning satisfaction in the association between effort-reward imbalance and academic burn-out.

Title 1	Effect	S. E.	P	LLCI	ULCI	Relative Effect Size
Total effect	0.9269	0.0786	0.0000	0.7726	1.0811	
Direct effects	0.4381	0.0718	0.0000	0.2972	0.5791	47.26%
Indirect effects	0.4887	0.0553	/	0.3869	0.6019	52.72%

**Table 4 behavsci-13-00028-t004:** Analysis of the moderating effect of resilience.

	Coeff	S. E.	*t*	*p*	95% CI
Constant	3.3452	0.0690	48.5139	0.0000	(3.2099, 3.4806)
Effort-Reward Imbalance Rate	−0.4148	0.0677	−6.1253	0.0000	(−0.5478, −0.2819)
Resilience	0.3976	0.0282	14.0971	0.0000	(0.3422, 0.4530)
Effort-Reward Imbalance Rate × Resilience	−0.2008	0.0857	−2.3420	0.0194	(−0.3690, −0.0325)

**Table 5 behavsci-13-00028-t005:** Mediating effects at different levels of resilience.

Item	Resilience		Effect	Boot SE	95% BootCI
A moderating mediating effect	M − 1 SD	2.7361	0.1919	0.0495	(0.1001, 0.2953)
	M	3.3486	0.2727	0.0493	(0.1778, 0.3688)
	M + 1 SD	3.9611	0.3535	0.0693	(0.2142, 0.4883)

## Data Availability

The data included in this study are available from the author upon request.

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
