# Peer review of "The Relationship between Effort-Reward Imbalance for Learning and Academic Burnout in Junior High School: A Moderated Mediation Model"

_behavsci, 2022, doi:10.3390/bs13010028_

Round 1
Reviewer 1 Report
Your work titled “The Relationship between Effort-Reward Imbalance for Learning and Academic Burnout in Junior High School: A Moderated Mediation Model” is an important manuscript about education and school. I congratulate you for raising such an important issue. In general, your work has been a good study in terms of theory, model and analysis. However, I have suggested some corrections. I have shared these corrections with you one by one below. I wish you success.
- Your abstract should have one or two sentences that are relevant to the topic and problem as your introductory sentence. Writing this sentence will allow readers to understand what the topic and problem are.
- “To explore the influence mechanism between effort-reward imbalance for learning and academic burnout among junior high school students”. What exactly does the sentence mean?
- In the summary part, I suggest you write in one sentence how the data was collected, which method was applied, which analyzes and programs were used.
-You can write some implications at the end of the summary.
You can write a research question at the end of the introduction. Likewise, the theory on which the study is based should be written here.
-Do you have any demographic questions?
- Two sample items should be written from the scales.
- Didn't you use control variables in the analysis?
- There is no data analyzes subheading under the Method heading. It must be written.
There's no point in having Gender in correlation analysis.
-The resources used in the discussion are insufficient and the discussion should be written separately for each hypothesis.
- The research question that will be written in the Introduction part should be evaluated in the last part of the discussion.
- Limitation title should be written separately.
- Conclusions and Implications title should be written separately.
- Manuscript contains English typos.
Author Response
Reviewer 1
Comments and Suggestions for Authors
Your work titled “The Relationship between Effort-Reward Imbalance for Learning and Academic Burnout in Junior High School: A Moderated Mediation Model” is an important manuscript about education and school. I congratulate you for raising such an important issue. In general, your work has been a good study in terms of theory, model and analysis. However, I have suggested some corrections. I have shared these corrections with you one by one below. I wish you success.
- Your abstract should have one or two sentences that are relevant to the topic and problem as your introductory sentence. Writing this sentence will allow readers to understand what the topic and problem are.
Response: Thank you for the valuable comments. We have added a paragraph that gives a brief introduction of the current research status in this field to the abstract and a corresponding description of the research theme and research questions to the paper.
- “To explore the influence mechanism between effort-reward imbalance for learning and academic burnout among junior high school students”. What exactly does the sentence mean?
Response: Thank you for the valuable comments. I apologize that we did not clearly convey the intended meaning. We changed the sentence to, “This study, from the perspective of learning satisfaction and resilience, intends to investigate how effort-reward imbalance affects academic burnout and reveal the influence of effort-reward imbalance on academic burnout. ”
- In the summary part, I suggest you write in one sentence how the data was collected, which method was applied, which analyzes and programs were used.
Response: Thank you for the valuable comments. I apologize that we did not include this part in this manuscript. We have supplemented it in the revised manuscript and added this passage to the Measure section.
-You can write some implications at the end of the summary.
Response: Thank you for the valuable comments. We added some implications at the end of the summary.
You can write a research question at the end of the introduction. Likewise, the theory on which the study is based should be written here.
Response: Thank you for the valuable comments. We have added research questions in the revised manuscript and supplemented all the theories in this manuscript.
-Do you have any demographic questions?
Response: Thank you for the valuable comments. I apologize that we did not administer demographic questions when we collected the data.
- Two sample items should be written from the scales.
Response: Thank you for the valuable comments. I apologize that we did not include this part in the manuscript. We supplemented the revised manuscript and added two samples to each scale.
- Didn't you use control variables in the analysis?
Response: Thank you for the valuable comments. I apologize that we did not use control variables in the data analysis process. In the revised document, we controlled for two variables, gender and grade, and reanalyzed the data.
- There is no data analyzes subheading under the Method heading. It must be written.
Response: Thank you for the valuable comments. I apologize that we did not properly format the manuscript so that the headings were in the same format as the text. In the revised manuscript, we revised the format and added some headings.
There's no point in having Gender in correlation analysis.
Response: Thank you for the valuable comments. We removed the gender variable from the relevant analysis in the revised manuscript.
-The resources used in the discussion are insufficient and the discussion should be written separately for each hypothesis.
Response: Thank you for the valuable comments. The discussion section in the revised manuscript is divided into three categories, and the limitations section of the research has been added to the original section. At the same time, we added related references in this part to make the discussion section more convincing.
- The research question that will be written in the Introduction part should be evaluated in the last part of the discussion.
Response: Thank you for the valuable comments. In the discussion section of the revised manuscript, we added the prospect of this research question and evaluated the research question accordingly.
- Limitation title should be written separately.
Response: Thank you for the valuable comments. Limitations and future expectations of this study have been included in the discussion section of the revised manuscript.
- Conclusions and Implications title should be written separately.
Response: Thank you for the careful review. We separated them in the revised manuscript.
- Manuscript contains English typos.
Response: Thank you for the careful review. After the manuscript was revised, we found a professional editing company to polish the revised version.
Reviewer 2 Report
The study is based on a social need and the theme is applicable to the secondary education level and to other higher levels such as professional training and university, I value the development of concepts such as resilience and the search for balance between effort-reward. It has been a success to adapt already validated scales and have a sample of 755 students
As an area for improvement, it could include qualitative data that shed light on the reasons for these results, although we understand that this can be done in future research and not in this one.
I consider the results found to be of great interest and the methodology carried out is correct.
I propose to add a small paragraph with the limitations found in the research, since they can help future researchers who want to delve into this subject.
I recommend doing a last reading of the document to avoid repetition of concepts and go to the essence of the content. Finally, I propose to include the prospective of the research between 1 and 5 years. Knowing what the follow-up or possible future application will be is of the utmost interest.
congratulations for the job
Author Response
Reviewer 2
Comments and Suggestions for Authors
The study is based on a social need and the theme is applicable to the secondary education level and to other higher levels such as professional training and university, I value the development of concepts such as resilience and the search for balance between effort-reward. It has been a success to adapt already validated scales and have a sample of 755 students
As an area for improvement, it could include qualitative data that shed light on the reasons for these results, although we understand that this can be done in future research and not in this one.
I consider the results found to be of great interest and the methodology carried out is correct.
I propose to add a small paragraph with the limitations found in the research, since they can help future researchers who want to delve into this subject.
Response: Thank you for the valuable comments. In the discussion section of the revised manuscript, we have added the limiting factor of the research question.
I recommend doing a last reading of the document to avoid repetition of concepts and go to the essence of the content. Finally, I propose to include the prospective of the research between 1 and 5 years. Knowing what the follow-up or possible future application will be is of the utmost interest.
Response: Thank you for the valuable comments. In the final part of the revised manuscript, we have added areas that may need to be explored in the future.

Reviewer 3 Report
I have no objections to the article. I propose to complete the scientific objective of the article in the introduction
Author Response
reviewer 3
Comments and Suggestions for Authors
I have no objections to the article. I propose to complete the scientific objective of the article in the introduction
Response: Thank you for taking time out of your busy schedule to review and confirm our manuscript. We thank the experts for the comments and suggestions. I have revised the entire paper according to the experts' opinions and further discussed the scientific objectives of the paper in the introduction.

Round 2
Reviewer 1 Report
- What is written under the heading of statistical analysis is not enough. How did you make Bootstrap? Which programs did you analyze? Write down the data analysis process. So where did you do the mediation and moderation analysis? How did you draw the graph (simple slope)? Please explain these details.
- The resources you use in the literature are not enough.
- Before the hypotheses, write the theory and literature that will underpin them, and then write your hypotheses based on them. So don't start directly with hypotheses.
- The title below is not correct. "4.2. Mediating effect of learning satisfaction on effort-reward imbalance and academic burnout". It could be: 4.2. Mediating effect of learning satisfaction in the relationship between effort-reward imbalance and academic burnout.
- Please review the entire text according to the warning above.
- I wanted you to write a research question in the introduction part and evaluate it in the last part of the discussion, but you did not write the question in the introduction part.
- I could not see your Conclusion and Implications title at the end of the article. It must be written.
Author Response
Comments and Suggestions for Authors
- What is written under the heading of statistical analysis is not enough. How did you make Bootstrap? Which programs did you analyze? Write down the data analysis process. So where did you do the mediation and moderation analysis? How did you draw the graph (simple slope)? Please explain these details.
Response: Thank you for the valuable comments. We have supplemented this in the statistical analysis section,for example “A sample size of 5000 was drawn by bootstrap method, confidence interval excluding 0 representing significant representative parameter, mediation effect analysis using model4 and regulated mediation effect analysis by model7.”
- The resources you use in the literature are not enough.
Response: Thank you for the valuable comments. In this revision, we have added to the relevant literature.
- Before the hypotheses, write the theory and literature that will underpin them, and then write your hypotheses based on them. So don't start directly with hypotheses.
Response: Thank you for the valuable comments. We adjusted for the location of the assumptions, while streamlining the articles and assumptions.
- The title below is not correct. "4.2. Mediating effect of learning satisfaction on effort-reward imbalance and academic burnout". It could be: 4.2. Mediating effect of learning satisfaction in the relationship between effort-reward imbalance and academic burnout.
Response: Thank you for the valuable comments. We have revised it in the article.
- Please review the entire text according to the warning above.
- I wanted you to write a research question in the introduction part and evaluate it in the last part of the discussion, but you did not write the question in the introduction part.
Response: Thank you for the valuable comments. In the revised article, we have added a research question in the introduction part and evaluated it in the last part of the discussion
- I could not see your Conclusion and Implications title at the end of the article. It must be written.
Response: Thank you for the valuable comments. In the revised article, we have added the conclusion and implications.

Round 3
Reviewer 1 Report
You need to correct the statements below.
3.3. The moderating effect of resilience on effort-reward imbalance and learning satisfaction
The moderating effect of resilience in the relationship between effort-reward imbalance and learning satisfaction
Table 3. Mediating effect of learning satisfaction on effort-reward imbalance and academic burn-out.
Mediating effect of learning satisfaction in the association between effort-reward imbalance and academic burn-out.